# A standard calibration method based on a symmetric resistance network matrix for galvanic logging instruments

Yongli Ji, Zhiqiang Li*, Jigen Xia, Jiajia Song

China Research Institute of Radiowave Propagation, Xinxiang, Henan, China

* lizhiqiang316@126.com

## Abstract

An increasing number of measurement electrodes have been designed to satisfy the demand for high-resolution detection using galvanic logging technology in complex formations. The forward modeling response analysis of logging tools has important guiding significance in the design of galvanic logging tools. Based on a three-dimensional finite element numerical simulation method, we established a forward model of galvanic multi-electrodes in a complex formation. We also designed a symmetrical resistance network model of the formation with equivalent resistance between two electrodes. A symmetrical resistance network was derived using the balanced bridge method. The asymmetrical admittance matrix was extended to a symmetrical extended admittance matrix to realize a convenient calculation of the equivalent symmetrical resistance network in complex formations. Verification of the microcolumn-focused logging tool, with nine electrodes in a simulated standard well, and an evaluation of the degree of invasion in an actual oil well indicate that this calibration method can improve the measurement accuracy of galvanic logging instruments.

## Introduction

Galvanic logging is the oldest and most commonly used logging technology. Ordinary resistivity logging technology was created in the 1920s, which is a type of non-aggregated electrode logging. The number of electrodes is small, and the response analysis is simple. However, this method is significantly influenced by the borehole and adjacent layers. Subsequently, various lateral logging technologies have been developed, with an improvement in the measurement accuracy by focusing on electrodes. Currently, the most commonly used tools in China are the HRLA (Schlumberger Company), HDLL array lateral (Atlas Company), and microcolumn focused logging tools [1–3]. Schlumberger's HRLA instrument has 16 electrodes that can adequately describe the anisotropy of the formation [4]. The microcolumn focusing logging tool consists of nine electrodes and uses a digitally synthesized focused method to improve the resolution of the formation resistivity. Yuan [5] proposed that an analysis of the response of these focused electrode logging tools can be performed by combining 3-D finite element theoretical simulations and scaled-down laboratory equivalent tests. A 3-D finite element method (FEM) simulation requires the establishment of complex formation models and tools of actual size. Neumann boundary conditions have been set for the surface of the instrument electrodes and

**Competing interests:** The authors have declared that no competing interests exist.

insulating ring. Non-uniform grids have been used for the surface of the instrument electrodes, with the use of formation fractures for encryption. Each calculation must solve a large sparse equation. To verify the correctness of the theoretical simulation, laboratory tests are required, which necessitate the development of logging instruments scaled-down proportionally and the establishment of different complex formation models. Mutual verification of the two can improve the practical measurement of the logging instrument precision. Barlai [6] proposed the use of an equivalent earth network to realize simulation and calibration simulations of logging tools and provided an equivalent full resistance network model, which can be used to simulate complex formations. However, they did not provide further details and a lack of verification. Combined with the 3-D FEM, this network can avoid laboratory-scale tests and improve the analysis of the influence of the tools. Zhang [7] proposed a transmission-line impedance network model with Dirichlet boundary conditions to simulate 3-D resistivity forward models. Yang [8, 9] achieved an equivalent simulation of the formation in axisymmetric coordinates by assuming that the casing well is a line electrode and the formation of a resistance network in 3-D direct current and electromagnetic wave logging of metal casing wells. Gerami [10] used Kirchhoff's current and voltage laws to calculate the potentials of all nodes in the wireframe model of an equivalent resistance network. Taking microcolumn-focused logging (MCFL) as an example, 9 electrodes require 36 resistors to simulate complex formations, whereas HRLA requires 120, therefore necessitating the development of a calculation method for realizing large resistance networks [3].

This study simulated the response of an MCFL based on the 3-D FEM, constructed various anisotropic formation models, and establishes a 3-D equivalent model of a large resistance network for various anisotropic formations. A balanced bridge circuit was used to derive a multi-electrode full resistance network, obtaining an extended admittance matrix with real symmetric matrix properties. Symmetrical equivalence of the 3-D asymmetric formations was achieved, easily yielding the admittance matrix between any electrode and the grounding electrode.

## Standard calibration method for MCFL

The MCFL was first proposed by the Schlumberger Technology Company in the early 1990s. It provides three original measurement curves $R_{B0}$, $R_{B1}$, and $R_{B2}$, at different detection depths, which reflect the resistivity of the flushing zone, mud cake resistivity, and mud cake thickness, respectively. The MCFL can accurately measure the radial change in shallow resistivity with a higher vertical resolution than that of microresistivity logging and microsphere focusing logging [11, 12]. The latest MCFL adopts the digital focusing method, which avoids the problem where equipotential hardware focusing cannot eliminate the residual voltage; it has been widely used in major oil fields in China. Fig 1 shows the electrode distribution and working principle of the MCFL [3].

In Fig 1, A0 is the main electrode; A1, A1′, B0, B1, and B2 are the emission electrodes; M and M′ are the monitoring electrodes; N is the potential reference electrode; and B is the loop electrode. The micro-column electrode plate is near the mud cake. Current values of button electrodes B0, B1, and B2 can measure the apparent resistivity at different radial depths of the formation. Using digital focusing, the equipotentials of A0, M, B0, B1, and B2 can be achieved, ensuring that the current flows radially to the flushing zone and returns to the reference electrode, N. The flushing resistivity corresponding to the different radial depths of B0, B1, and B2 can be calculated as follows:

$$R_{Bi} = k_i \frac{U_{MN}}{I_{Bi}}, \tag{1}$$

where, $i = 1,2,3$, $k_i$ is the calibration coefficient of the button electrode, Bi, $R_{Bi}$ is the apparent

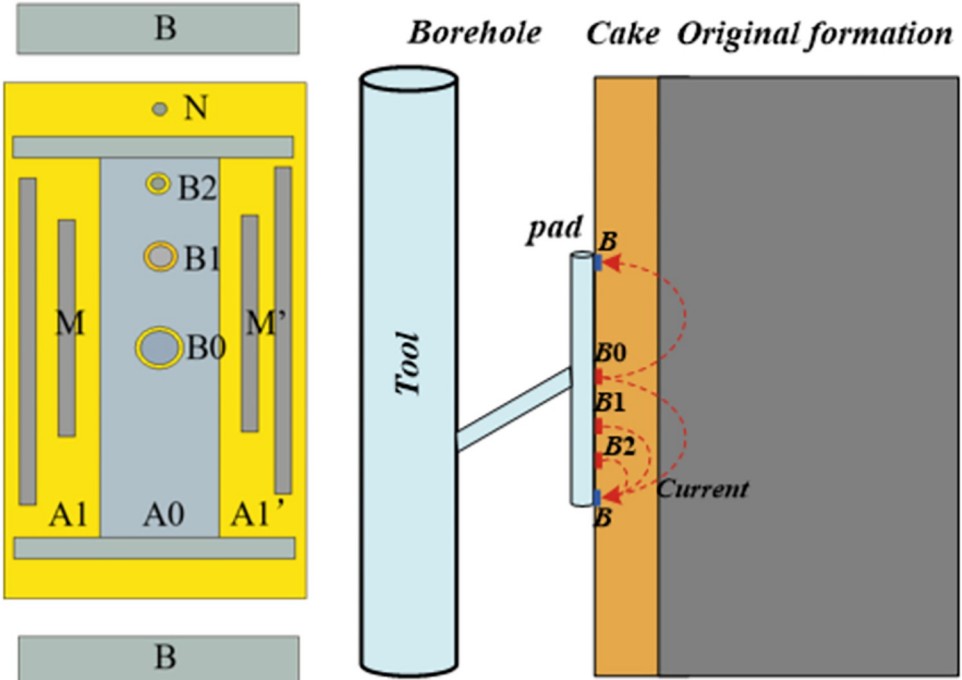

**Fig 1. Schematic diagram of MCFL tool.**

resistivity value of different zones, $I_{Bi}$ is the current value (unit A), and $U_{MN}$ is the potential difference between monitoring electrode, M and reference electrode, N (unit V).

The radial detection depth and invasion effect of the MCFL can be expressed by the pseudo-geometric factor, $J$:

$$J = \frac{R_{Bi} - R_t}{R_{xo} - R_t},\tag{2}$$

where, $R_t$ represents the original formation resistivity (unit Ω·m). $R_{xo}$ represents the intrusion resistivity (unit Ω·m).

Here, $R_{xo}$ can be inversed using Eq (2) [3]. The calibration coefficient, $k_i$ is typically measured using a sophisticated circuit. However, an increasing number of electrodes yields mor edifficulty in designing an appropriate circuit. To avoid various calibration circuit designs for different galvanic logging instruments, we introduce a standard calibration method based on an FEM simulation and a symmetric resistance network matrix. First, a simulation model was determined and meshed according to a real logging tool. Second, the FEM was used to calculate the potential difference between the electrodes. Furthermore, a symmetric resistance network matrix with unknown resistor values was designed. Then, an inversion method was introduced to obtain the values of all of the resistor. Finally, a real resistance circuit was constructed and connected to the MCFL. Here, $k_i$ is the ratio of the model MCFL to the measured MCFL. Fig 2 shows the workflow of this study.

According to the FEM, the forward calculation of the MCFL response can be transformed into the problem of determining the extreme value of the function shown in Eq (3) [13]:

$$F(\Phi) = \frac{1}{2} \iiint_V \sum_{i,j=1}^{3} \sigma_{\xi_i \xi_j} \frac{\partial \Phi}{\partial \xi_i} \frac{\partial \Phi}{\partial \xi_j} - \sum_E I_E U_E dV,\tag{3}$$

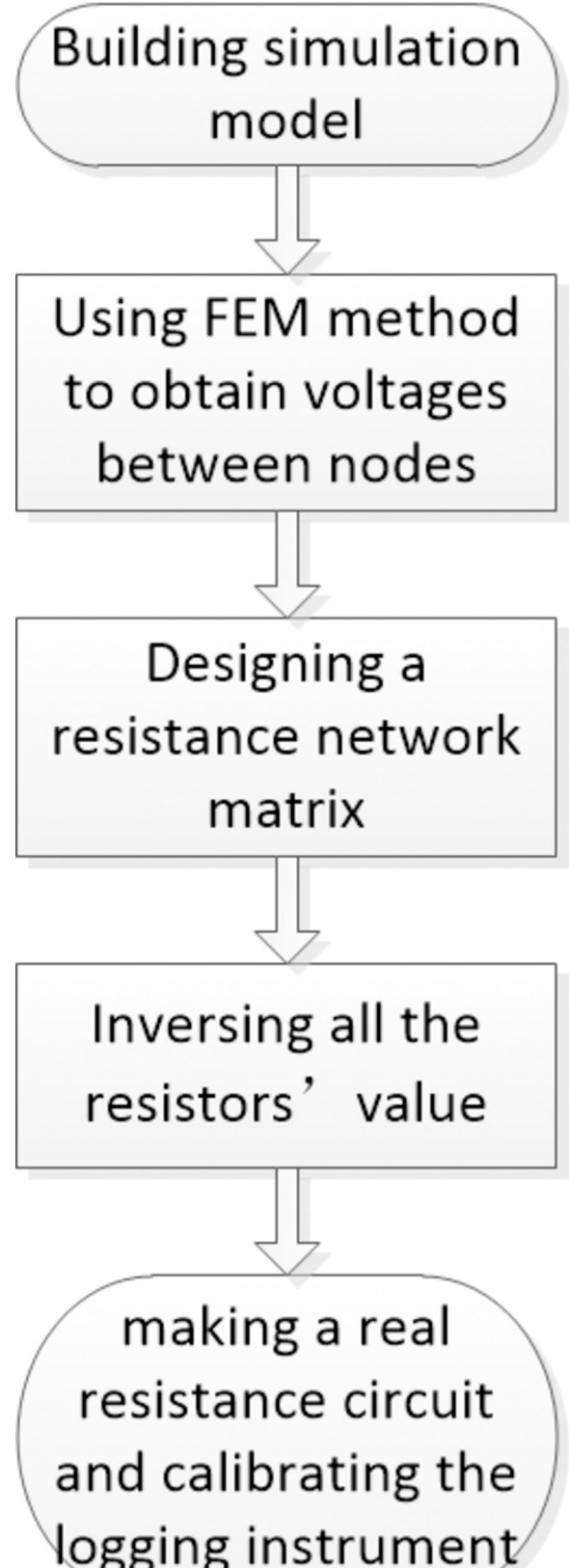

**Fig 2. Workflow of the standard calibration for galvanic logging instruments.**

where, $\Phi$ is the node's potential, $\xi_i$ is the position of the three vertices of the triangle grid (unit V), $I_E$ and $U_E$ is the current and voltage, respectively (unit A and V), and $\sigma_{\xi_i\xi_j}$ is the conductivity of the mesh grid (unit S/m).

In the forward calculation model, two types of boundary conditions exist: Dirichlet and Nemmann. At an infinitely distant formation boundary, the potential satisfied the Dirichlet boundary condition:

$$\begin{cases} \Phi|_{z\to\infty} = 0 \\ \Phi|_{\sqrt{x^2+y^2}\to\infty} = 0 \end{cases} \tag{4}$$

The Neumann boundary condition was met on the surface of the instrument electrode and insulation ring of the instrument:

$$\begin{cases} (\sigma\nabla\Phi)\cdot e_n|_{\Gamma_I} = 0 \\ (\sigma\nabla\Phi)\cdot e_n|_{\Gamma_C} = j_s \end{cases} \tag{5}$$

where, $e_n$ is the unit normal vector, $j_s$ is the current density (unit A/m$^2$), $\Gamma_I$ and $\Gamma_C$ are the boundaries.

The FEM solution included area and function discretization. When performing FEM segmentation, dense nodes were set on the electrode system; sparse nodes were set in other areas. The value of the bit function for each element node after segmentation was approximated using an appropriate interpolation method. The function became a quadratic form containing the upper function of each node. When the functional reached its minimum value, the potential distribution at each node was an approximate solution to the actual electromagnetic field. Therefore, grid discretization of the stratigraphic model and instruments was performed. The derivative of Eq (3) was obtained as follows:

$$\frac{\partial F(\Phi)}{\partial\Phi} = K\tilde{\Phi} - b, \tag{6}$$

where, $K$ denotes the stiffness matrix, $\tilde{\Phi}$ is a large sparse matrix composed of the potentials of all of the nodes in the mesh grid, and $b$ is a vector of the generated source.

By setting Eq (6) to zero, we can calculate the electric field generated by any electrode (Fig 1) as the emission electrode in the formation.

## Symmetrical resistance network

The MCFL tool shown in Fig 1 uses a plate pushing method. At this time, the formation does not have axial symmetry and can only be analyzed using 3-D numerical simulation methods. The FEM forward mesh division was directly proportional to the complexity of the formation, which affected the efficiency of the forward calculation. Using an equivalent resistance network to simulate complex formations, the measurement of logging tools can be easily calibrated, with improvements to the inversion accuracy of the formations. For the MCFL tool, a network consisting of 36 resistors, as shown in Fig 3, was required for the nine electrodes to be equivalent.

For simplicity, taking a 4-node resistor network as an example, as shown in Fig 4, the resistance was represented in admittance form for the subsequent derivation. The number of nodes and branches can be arbitrarily selected; the direction of current can also be arbitrarily selected. The solution method for the equivalent admittance and node potential was based on the principle of bridge balance [14].

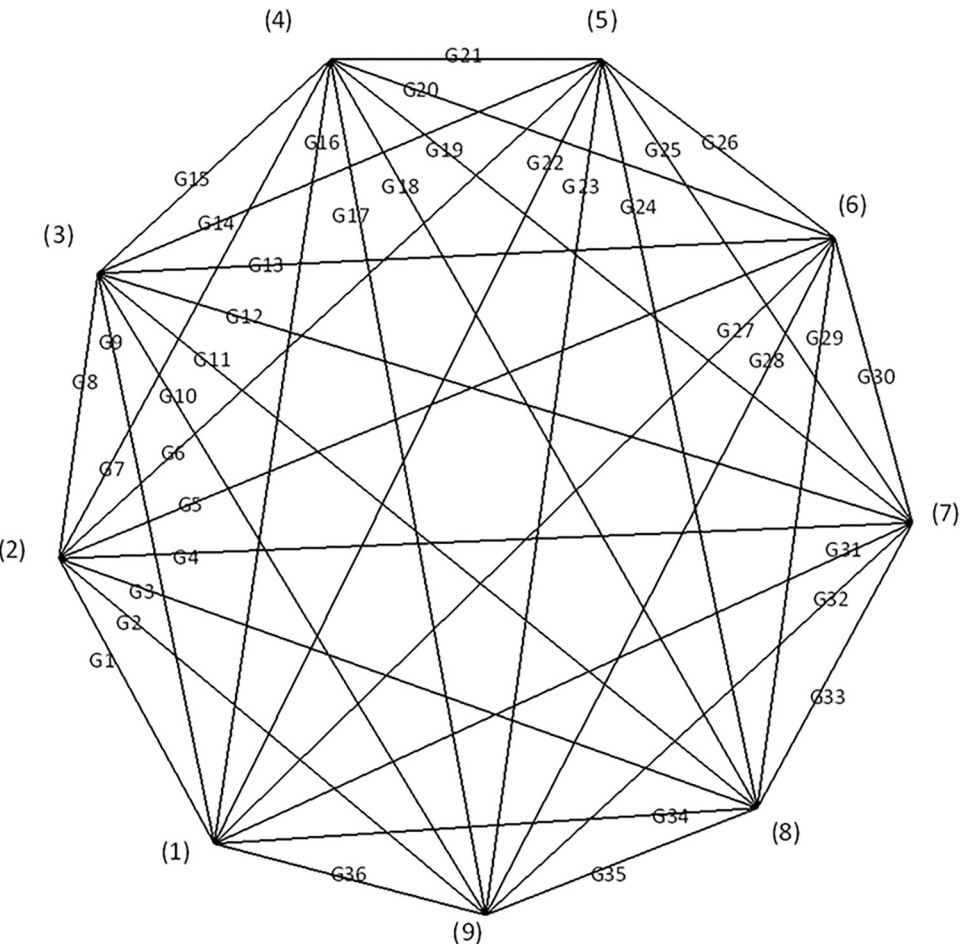

**Fig 3. Equivalent resistor network.**

The node-branch correlation matrix, A, conductance matrix, G, and source vector, $e_s$, $i_s$, can be expressed as follows:

$$A = \begin{bmatrix} -1 & 1 & 0 & 1 & 0 & 0 \\ 0 & -1 & 1 & 0 & 0 & 1 \\ 0 & 0 & 0 & -1 & -1 & -1 \end{bmatrix} \text{ and} \tag{7}$$

$$G = \begin{bmatrix} G_1 & & & & & \\ & G_2 & & & & \\ & & G_3 & & & \\ & & & G_4 & & \\ & & & & G_5 & \\ & & & & & G_6 \end{bmatrix}. \tag{8}$$

To obtain the source vector, the power supply can be transformed into an ideal current source, $i_s$, and a parallel circuit with conductivity G, with the direction taken as the same as

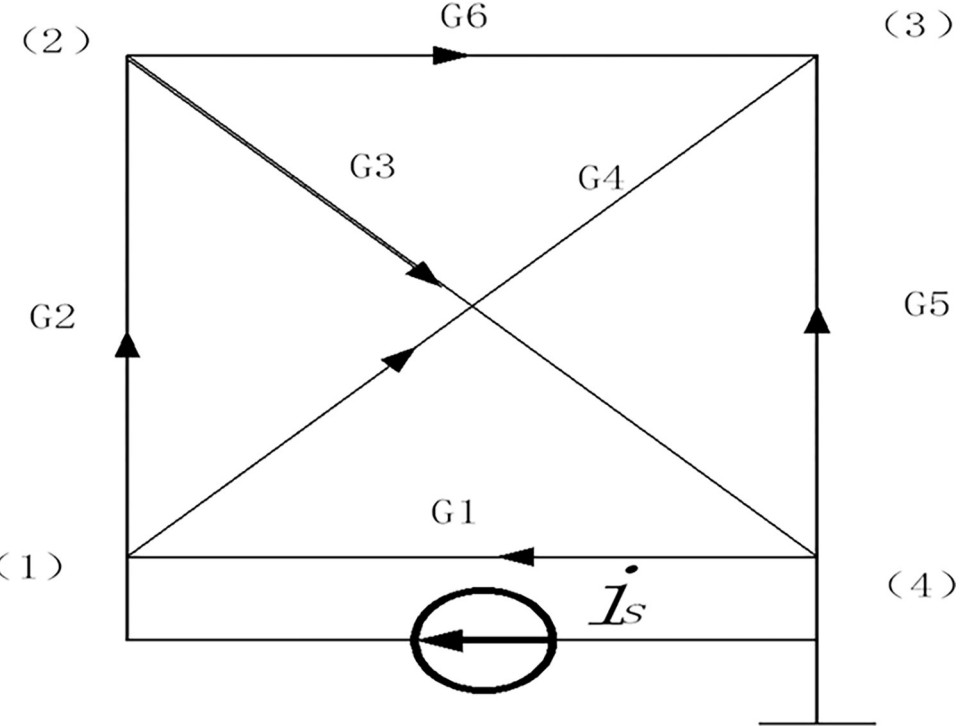

**Fig 4. Quadrilateral resistance network.**

branch 1. Therefore, $i_s$ and $e_s$ can be written as follows:

$$i_s = \begin{bmatrix} i_{s1} & 0 & 0 & 0 & 0 & 0 \end{bmatrix}^T \text{ and} \tag{9}$$

$$e_s = \begin{bmatrix} 0 & 0 & 0 & 0 & 0 & 0 \end{bmatrix}^T. \tag{10}$$

Based on the above equations, the node admittance matrix, $Y_n$ and node source current vector, $i_{ns}$ can be calculated as follows:

$$Y_n = AGA^T \text{ and} \tag{11}$$

$$i_{ns} = AGe_s - Ai_s. \tag{12}$$

Substituting Eqs (7)–(10) into Eqs (11) and (12), we obtained the following:

$$Y_n = \begin{bmatrix} G_1 + G_2 + G_4 & -G_2 & -G_4 \\ -G_2 & G_2 + G_3 + G_6 & -G_6 \\ -G_4 & -G_6 & G_4 + G_5 + G_6 \end{bmatrix} \text{ and} \tag{13}$$

$$i_{ns} = \begin{bmatrix} i_{s1} & 0 & 0 \end{bmatrix}^T. \tag{14}$$

The potentials of the electrodes are as follows:

$$Y_n \varphi = i_{ns} \tag{15}$$

where, $\varphi = \begin{bmatrix} \varphi_1 & \varphi_2 & \varphi_3 \end{bmatrix}^T$ is the vector of the potentials.

The derivation process of the above equations clarifies the relationship between each resistance of the full resistance network, node potential, and current source. The current source can be set between the other two nodes to perform the same steps, according to Eqs (7)–(15).

Based on Eq (15), when the current source is known, the potential of each electrode can be solved according to the simulated formation resistivity information, followed by calculating the resistance value of each resistor. The number of resistors in Eq (15) is significantly larger than the number of nodes. After the position of each current source is determined, $N-1$ equations can be formed. At least $M/(N-1)$ different current source positions are required to obtain a unique solution. Assuming that P electrode pairs are calculated, $P \geq M/(N-1)$, the following equation can be obtained:

$$ZY = I, \tag{16}$$

where, Z is the coefficient matrix of Y, which is obtained from P different current source positions, such that the order of Z is PM×M, $Y = \begin{bmatrix} G_1 & G_2 & \cdots & G_M \end{bmatrix}^T$ is the admittance vector, and $I = \begin{bmatrix} i_{ns} & \cdots & i_{ns} \end{bmatrix}^T$ is the new vector consisting of $i_{ns}$.

The value of each resistor can be obtained by inverting Eq (16):

$$Y = (Z^T Z)^{-1} Z^T I. \tag{17}$$

The specific algorithm is as follows:

1. Add a current source between two nodes to obtain the potential of all nodes.

2. Derive the corresponding admittance matrix from the Eq (13).

3. Substitute into Eq (15) to obtain the coefficients of each resistor and write them in matrix form.

4. Repeat the above three steps until the system of equations is well or over-determined.

5. Finally, use Eq (17) to calculate the value of each resistor.

Based on the above algorithm, the most time-consuming step is generating an admittance matrix. By observing the composition of each admittance matrix, the following matrix was obtained:

$$G = \begin{bmatrix} G_1 + G_2 + G_4 & -G_2 & -G_4 & -G_1 \\ -G_2 & G_2 + G_3 + G_6 & -G_6 & -G_3 \\ -G_4 & -G_6 & G_4 + G_5 + G_6 & -G_5 \\ -G_1 & -G_3 & -G_5 & G_1 + G_3 + G_5 \end{bmatrix}. \tag{18}$$

Eq (18) is referred to as the extended admittance matrix, which is a symmetric matrix, and can be written quickly according to Fig 4. The diagonal line represents the sum of the admittances between nodes. The first row shows the different admittances of node (1); the second row shows the different admittances of node (2); the third row shows the different admittances of node (3); and the fourth row shows the different admittances of node 4. The admittance after the diagonal elements in each row is filled in clockwise with a negative sign. We note that the nodes and admittance numbers can be written arbitrarily if they are sequentially generated. The admittance matrix corresponding to each node can be obtained using the following steps.

1. Nodes (1) and (4) are connected to form a branch; if node (4) is a loop, then G(1:3,1:3) is an admittance matrix picked from Eq (17) by the index of rows and columns.

2. Nodes (2) and (1) are connected to form a branch; if node (1) is a loop, then G(2:4,2:4) is an admittance matrix.

3. Nodes (3) and (2) are connected to form a branch; if node (2) is a loop, then G(1 3 4,1 3 4) is an admittance matrix.

4. Nodes (4) and (3) are connected to form a branch; if node (3) is a loop, then G(1 2 4, 1 2 4) is an admittance matrix.

Therefore, the symmetric resistance network (SRN) algorithm for calculating the value of each resistor can be modified as follows:

1. Write the extended admittance matrix according to the resistor network number.

2. Add a current source between two nodes to obtain the potential of all nodes.

3. Take out the corresponding admittance matrix from step 1).

4. Substitute it into Eq (15) to obtain the coefficients of each resistor and write them in matrix form.

5. Repeat the above three steps until the system of equations is well or over-determined.

6. Finally, use Eq (17) to calculate the resistance value.

For the equivalent resistance network of the nine electrodes shown in Fig 3, the extended admittance matrix is as follows:

$$
G = \begin{bmatrix}
D_1 & -G_1 & -G_9 & -G_{16} & -G_{22} & -G_{27} & -G_{31} & -G_{34} & -G_{36} \\
-G_1 & D_2 & -G_8 & -G_7 & -G_6 & -G_5 & -G_4 & -G_3 & -G_2 \\
-G_9 & -G_8 & D_3 & -G_{15} & -G_{14} & -G_{13} & -G_{12} & -G_{11} & -G_{10} \\
-G_{16} & -G_7 & -G_{15} & D_4 & -G_{21} & -G_{20} & -G_{19} & -G_{18} & -G_{17} \\
-G_{22} & -G_6 & -G_{14} & -G_{21} & D_5 & -G_{26} & -G_{25} & -G_{24} & -G_{23} \\
-G_{27} & -G_5 & -G_{13} & -G_{20} & -G_{26} & D_6 & -G_{30} & -G_{29} & -G_{28} \\
-G_{31} & -G_4 & -G_{12} & -G_{19} & -G_{25} & -G_{30} & D_7 & -G_{33} & -G_{32} \\
-G_{34} & -G_3 & -G_{11} & -G_{18} & -G_{24} & -G_{29} & -G_{33} & D_8 & -G_{35} \\
-G_{36} & -G_2 & -G_{10} & -G_{17} & -G_{23} & -G_{28} & -G_{32} & -G_{35} & D_9
\end{bmatrix},
\tag{19}
$$

where,

$$
D_1 = G_1 + G_9 + G_{16} + G_{22} + G_{27} + G_{31} + G_{34} + G_{36},
\tag{20}
$$

$$
D_2 = \sum_{i=1}^{8} G_i,
\tag{21}
$$

$$
D_3 = \sum_{i=8}^{15} G_i,
\tag{22}
$$

$$
D_4 = G_7 + \sum_{i=15}^{21} G_i,
\tag{23}
$$

$$
D_5 = G_6 + G_{14} + \sum_{i=21}^{26} G_i,
\tag{24}
$$

$$D_6 = G_5 + G_{13} + G_{20} + \sum_{i=26}^{30} G_i, \tag{25}$$

$$D_7 = G_4 + G_{12} + G_{19} + G_{25} + \sum_{i=30}^{33} G_i, \tag{26}$$

$$D_8 = G_3 + G_{11} + G_{18} + G_{24} + G_{29} + \sum_{i=33}^{35} G_i, \text{ and} \tag{27}$$

$$D_9 = G_2 + G_{10} + G_{17} + G_{23} + G_{28} + G_{32} + \sum_{i=35}^{36} G_i. \tag{28}$$

## Verification

The following parameters were used to simulate the response of the MCFL based on the FEM, as listed in Table 1.

$R_m$ is the well fluid resistivity (unit Ω·m); $R_{mc}$ is the mud cake resistivity (unit Ω·m); $H_{xo}$ and $H_{mc}$ are the thickness of the intrusion zone and mud cake, respectively (unit inches).

The parameters in Table 1 were used to calculate the potentials of all of the electrodes. Tables 2–10 list the calculation results.

Substituting the potentials in Tables 2–9 and the admittance matrix into Eq (15), finally combining them into Eq (16), 36 resistance values were obtained using Eq (17), as listed in Table 10.

The resistance network listed in Table 10 is symmetrical, and the theoretical calibration coefficient is 1. A real MCFL instrument can then be connected to this network to perform calibration.

The network can be used for the quantitative calibration of direct current logging and to establish a quantitative relationship between the reading and physical properties of the complex formation, such that the measured value of the instrument directly reflects the physical property value.

**Table 1. 3-D inhomogeneous formation model parameters.**

| Model name | | Model parameter |
|---|---|---|
| Electrode plate | length | 250 cm |
| | diameter | 89 cm |
| | B0 | 19 cm |
| | B1 | 14 cm |
| | B2 | 9 cm |
| | N | 5 cm |
| Mesh size | length | 30 m |
| | width | 30 m |
| | height | 30 m |
| current | | 1 A |
| Formation | $R_t$ | 10 Ω·m |
| | $R_m$ | 1 Ω·m |
| | $R_{xo}$ | 1 Ω·m |
| | $H_{xo}$ | 5 cm |
| | $R_{mc}$ | 1 Ω·m |
| | $H_{mc}$ | ¼ in |

**Table 2. D is the loop.**

| B−D | A0−D | M−D | A1−D | B0−D | B1−D | B2−D | N−D | |
|---|---|---|---|---|---|---|---|---|
| 0 | 0 | 0 | 0 | 0 | 0 | 0 | 0 | D |
| 8.9933 | 7.9354 | 7.9669 | 8.0405 | 7.9211 | 7.9216 | 7.9261 | 8.2352 | B |
| 7.9354 | 13.4136 | 12.3874 | 11.588 | 13.3621 | 13.3571 | 13.3382 | 11.0277 | A0 |
| 7.9669 | 12.3874 | 15.8369 | 12.4527 | 12.37 | 12.3503 | 12.3335 | 10.4486 | M |
| 8.0405 | 11.588 | 12.4527 | 13.6203 | 11.5682 | 11.5591 | 11.5506 | 10.0145 | A1 |
| 7.9211 | 13.3621 | 12.37 | 11.5682 | 20.0455 | 13.3077 | 13.2878 | 10.9927 | B0 |
| 7.9216 | 13.3571 | 12.3503 | 11.5591 | 13.3077 | 26.5534 | 14.9249 | 11.1023 | B1 |
| 7.9261 | 13.3382 | 12.3335 | 11.5506 | 13.2878 | 14.9249 | 25.863 | 11.5335 | B2 |
| 8.2352 | 11.0277 | 10.4486 | 10.0145 | 10.9927 | 11.1023 | 11.5335 | 61.3576 | N |

**Table 3. B is the loop.**

| A0−B | M−B | A1−B | B0−B | B1−B | B2−B | N−B | |
|---|---|---|---|---|---|---|---|
| 1.0579 | 1.0264 | 0.9528 | 1.0722 | 1.0717 | 1.0672 | 0.7582 | D |
| 0 | 0 | 0 | 0 | 0 | 0 | 0 | B |
| 6.5361 | 5.4784 | 4.6054 | 6.4988 | 6.4933 | 6.47 | 3.8505 | A0 |
| 5.4785 | 8.8964 | 5.4386 | 5.4754 | 5.4551 | 5.4339 | 3.24 | M |
| 4.6054 | 5.4385 | 6.5326 | 4.5999 | 4.5902 | 4.5773 | 2.7321 | A1 |
| 6.4989 | 5.4754 | 4.6 | 13.1967 | 6.4583 | 6.4339 | 3.8298 | B0 |
| 6.4934 | 5.4551 | 4.5903 | 6.4583 | 19.7035 | 8.0705 | 3.9389 | B1 |
| 6.47 | 5.4338 | 4.5773 | 6.4339 | 8.0705 | 19.0041 | 4.3656 | B2 |
| 3.8505 | 3.2399 | 2.7321 | 3.8298 | 3.9389 | 4.3655 | 53.8807 | N |

**Table 4. A0 is the loop.**

| M−A0 | A1−A0 | B0−A0 | B1−A0 | B2−A0 | N−A0 | |
|---|---|---|---|---|---|---|
| 1.0262 | 1.8256 | 0.0515 | 0.0565 | 0.0754 | 2.3859 | D |
| 1.0576 | 1.9307 | 0.0371 | 0.0427 | 0.0661 | 2.6856 | B |
| 0 | 0 | 0 | 0 | 0 | 0 | A0 |
| 4.4756 | 1.8908 | 0.0341 | 0.0194 | 0.0215 | 0.4471 | M |
| 1.8909 | 3.8579 | 0.0317 | 0.0276 | 0.038 | 0.8123 | A1 |
| 0.0341 | 0.0317 | 6.7349 | 0.0021 | 0.0011 | 0.0165 | B0 |
| 0.0194 | 0.0276 | 0.0021 | 13.2528 | 1.6432 | 0.1311 | B1 |
| 0.0215 | 0.038 | 0.0011 | 1.6432 | 12.6001 | 0.5812 | B2 |
| 0.4471 | 0.8124 | 0.0165 | 0.1311 | 0.5812 | 52.7158 | N |

**Table 5. M is the loop.**

| A1−M | B0−M | B1−M | B2−M | N−M | |
|---|---|---|---|---|---|
| 3.3842 | 3.4669 | 3.4866 | 3.5034 | 5.3883 | D |
| 3.4579 | 3.4211 | 3.4413 | 3.4627 | 5.6566 | B |
| 2.5848 | 4.4415 | 4.4562 | 4.4542 | 4.0286 | A0 |
| 0 | 0 | 0 | 0 | 0 | M |
| 4.5519 | 2.5824 | 2.593 | 2.6013 | 2.9501 | A1 |
| 2.5824 | 11.1424 | 4.4242 | 4.4212 | 4.011 | B0 |
| 2.593 | 4.4243 | 17.6897 | 6.078 | 4.1403 | B1 |
| 2.6013 | 4.4212 | 6.078 | 17.0328 | 4.5883 | B2 |
| 2.95 | 4.011 | 4.1403 | 4.5883 | 56.2973 | N |

**Table 6. A1 is the loop.**

| B0−A1 | B1−A1 | B2−A1 | N−A1 | |
|---|---|---|---|---|
| 2.0521 | 2.0613 | 2.0697 | 3.6059 | D |
| 1.9326 | 1.9424 | 1.9553 | 3.8005 | B |
| 3.8262 | 3.8304 | 3.8199 | 3.0456 | A0 |
| 1.9695 | 1.9589 | 1.9506 | 1.6018 | M |
| 0 | 0 | 0 | 0 | A1 |
| 10.5294 | 3.8007 | 3.7893 | 3.0304 | B0 |
| 3.8007 | 17.0556 | 5.4355 | 3.1491 | B1 |
| 3.7893 | 5.4356 | 16.3821 | 3.5888 | B2 |
| 3.0303 | 3.1491 | 3.5887 | 54.9491 | N |

**Table 7. B0 is the loop.**

| B1−B0 | B2−B0 | N−B0 | |
|---|---|---|---|
| 6.7379 | 6.7577 | 9.0528 | D |
| 6.7384 | 6.7628 | 9.3669 | B |
| 6.7329 | 6.7338 | 6.7184 | A0 |
| 6.7182 | 6.7212 | 7.1314 | M |
| 6.7288 | 6.7401 | 7.499 | A1 |
| 0 | 0 | 0 | B0 |
| 19.9836 | 8.375 | 6.8474 | B1 |
| 8.375 | 19.3329 | 7.2985 | B2 |
| 6.8475 | 7.2985 | 59.4177 | N |

**Table 8. B1 is the loop.**

| B2−B1 | N−B1 | |
|---|---|---|
| 11.6285 | 15.4511 | D |
| 11.633 | 15.7647 | B |
| 11.6097 | 13.1217 | A0 |
| 11.6117 | 13.5494 | M |
| 11.62 | 13.9065 | A1 |
| 11.6086 | 13.1361 | B0 |
| 0 | 0 | B1 |
| 22.5666 | 12.0597 | B2 |
| 12.0597 | 65.7064 | N |

**Table 9. B2 is the loop.**

| N−B2 | |
|---|---|
| 14.3295 | D |
| 14.6385 | B |
| 12.019 | A0 |
| 12.4446 | M |
| 12.7934 | A1 |
| 12.0344 | B0 |
| 10.5069 | B1 |
| 0 | B2 |
| 64.1536 | N |

**Table 10. Resistance values of the symmetrical resistance network.**

|    | B | A0 | M | A1 | B0 | B1 | B2 | N | D |
|----|----|----|----|----|----|----|----|----|----|
| B | 0 | 1185 | 16407 | 441362 | 12229 | 68276 | 96 | 130 | 3421 |
| A0 | 1185 | 0 | 100 | 247674 | 2623 | 37534 | 14 | 5807 | 7296 |
| M | 16407 | 100 | 0 | 53446 | 4301 | 9320 | 15 | 6119 | 6690 |
| A1 | 441362 | 247674 | 53446 | 0 | 1803 | 1323 | 7 | 3443 | 3146 |
| B0 | 12229 | 2623 | 4301 | 1803 | 0 | 7 | 7 | 12 | 153 |
| B1 | 68276 | 37534 | 9320 | 1323 | 7 | 0 | 7 | 219 | 779 |
| B2 | 96 | 14 | 15 | 7 | 7 | 7 | 0 | 14 | 84 |
| N | 130 | 5807 | 6119 | 3443 | 12 | 219 | 14 | 0 | 11 |
| D | 3421 | 7296 | 6690 | 3146 | 153 | 779 | 84 | 11 | 0 |

We compared and verified the results obtained by the MCFL instrument using actual calibrated and uncalibrated logging data, as well as the formation curve of a real well, as shown Fig 5. At 2224 to 2226 m, the calibrated logging curve followed the same pattern as the array induction logging curve, while there was a significant difference in the uncalibrated logging curve. From 2235–2240 m, the calibrated logging curve was closer to the array induction logging curve, whereas the uncalibrated logging results were significantly lower.

Fig 6 shows the oil and water layer evaluation results for an actual well in China. The conventional nine logging curves, array induction logging curves, high-resolution dual lateral

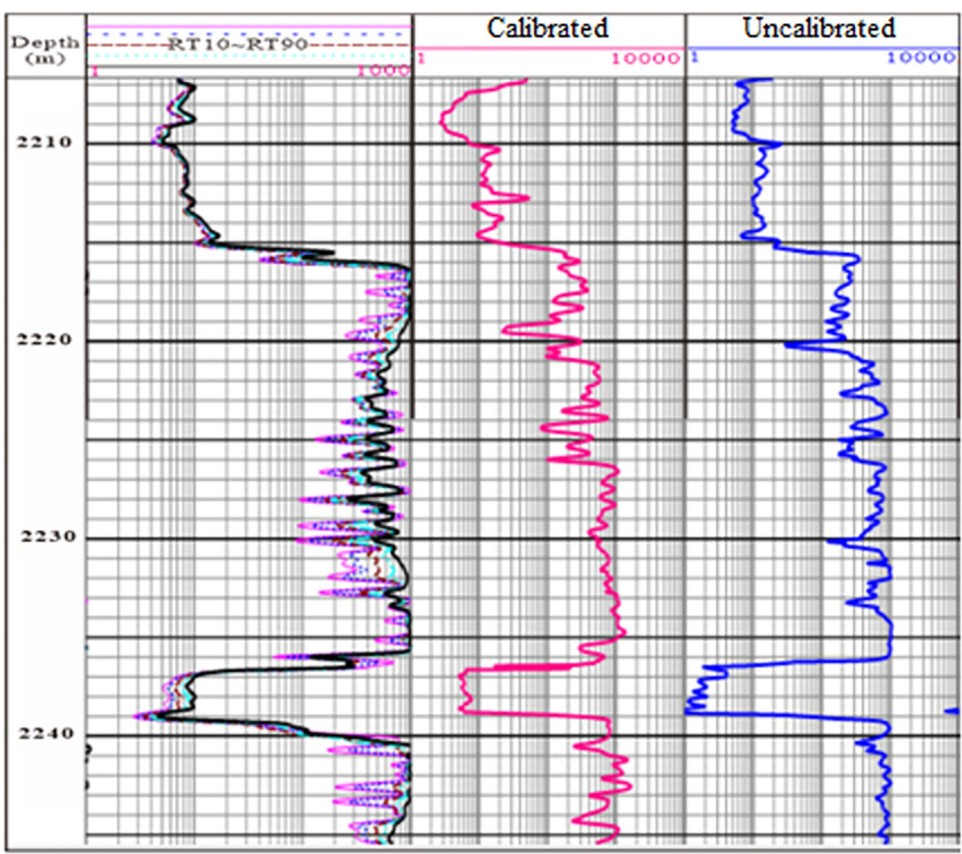

**Fig 5. Comparison of the logging results.**

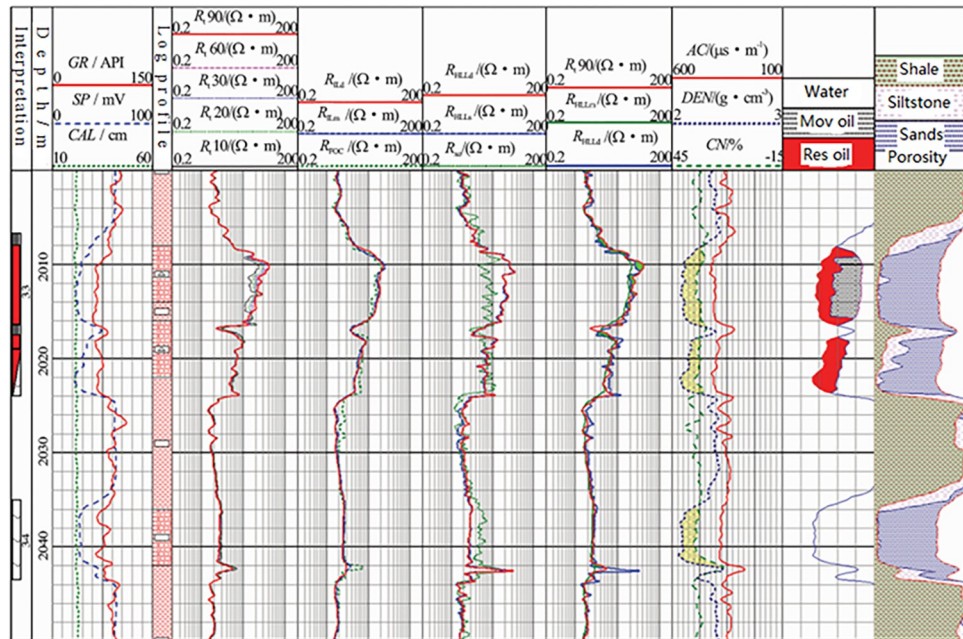

**Fig 6. Oil and water layer evaluation result of a real well in China.**

curves, and intrusion zone curve after precise MCFL calibration were measured in this well section. The section exhibited low invasion characteristics in the oil layer and high invasion characteristics in the water layer. Daily oil production from 2008.1–2011 m was 16.7 t, with a water content of 9%. This indicates the accuracy of the intrusion zone curve after MCFL's calibration.

## Discussion

The calibration method for galvanic logging instruments usually involves designing a calibration circuit that is easy to implement when there are a few electrodes. However, as the number of electrodes increases, the design of the calibration circuit becomes more complex. The simplest method is to make use of all resistors between each two electrodes to form a symmetric calibration network. Subsequently, the difficulty changes from the design of the calibration circuit to the solution of all of the calibration resistances. Previous studies focus on a method for calculating the equivalent resistance of a resistance network. The symmetry feature of our equivalent admittance matrix makes it easier to calculate each resistance from the equivalent resistance. A symmetrical full resistance calibration network can satisfy the calibration requirements of galvanic logging instruments in complex formations.

## Conclusions

As the oldest logging method, galvanic logging requires higher-accuracy measurements in more complex geological environments. Instrument design is becoming increasingly complex, and multiple-electrodes pose new challenges to instrument design calculation methods. Based on the working principle of the new 9-electrode MCFL tool, an equivalent symmetrical resistance network for complex formation was proposed and verified for galvanic logging instruments. It has a symmetric structure and simplifies the calculation and design of the calibration of galvanic logging instruments. This method also provides a method for solving the forward

problem of the equivalent resistance of complex resistance networks, as well as the inverse problem of solving every resistance.

## Supporting information

**S1 File. Some necessary data in Figs 5 and 6.**
(TXT)

## Author Contributions

**Conceptualization:** Zhiqiang Li.

**Data curation:** Jiajia Song.

**Formal analysis:** Yongli Ji.

**Investigation:** Yongli Ji.

**Methodology:** Yongli Ji.

**Resources:** Yongli Ji.

**Validation:** Jigen Xia.

**Writing – original draft:** Yongli Ji.

**Writing – review & editing:** Yongli Ji.

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
