## [Decision Letter · Decision Letter 0]

18 Dec 2023

PONE-D-23-37666A symmetric resistance network matrix calibration method for galvanic loggingPLOS ONE

Dear Dr. ji,

Thank you for submitting your manuscript to PLOS ONE. After careful consideration, we feel that it has merit but does not fully meet PLOS ONE’s publication criteria as it currently stands. Therefore, we invite you to submit a revised version of the manuscript that addresses the points raised during the review process.

We look forward to receiving your revised manuscript.

Kind regards,

Khalil Abdelrazek Khalil, Ph.D.

Academic Editor

PLOS ONE

Journal Requirements:

This work is supported by the national key R & D plan, Project number: 2019YFC0605101.

5. We note that your Data Availability Statement is currently as follows: All relevant data are within the manuscript and its Supporting Information files.

6. Please amend either the abstract on the online submission form (via Edit Submission) or the abstract in the manuscript so that they are identical.

Reviewers' comments:

Reviewer's Responses to Questions

**Comments to the Author**

1. Is the manuscript technically sound, and do the data support the conclusions?

Reviewer #1: Yes

Reviewer #2: Yes

2. Has the statistical analysis been performed appropriately and rigorously? 

Reviewer #1: Yes

Reviewer #2: N/A

3. Have the authors made all data underlying the findings in their manuscript fully available?

Reviewer #1: Yes

Reviewer #2: Yes

4. Is the manuscript presented in an intelligible fashion and written in standard English?

Reviewer #1: Yes

Reviewer #2: Yes

5. Review Comments to the Author

Reviewer #1: Nice work. The resistance network matrix is a powerful tool for the calibration of laterolog tool. By joint using the FEM and the resistance network, the necessary coefficients for the matrix is determined. These predetermined values then can be directly used to increase the calculation speed. I would like recommend a minor revision.

1 The sentences and grammar should be polished and modified by a native speaker. The current version is a little bit hard to understand;

2 It is better to give a workflow of the calibration method and describe the relationship between FEM and the resistance matrix method;

3 Please validate the accuracy of the new method in a layered medium.

Reviewer #2: 1. It was well-written. However, the abstract shall be improved. Please include the important result in the abstract.

2. All the variables, e.g., RB0, RB1, k0, Rx0, is1, es, etc., must be rewritten. They should be written in "equation" format.

3. All the mathematical expressions also should be rewritten in "equation" format.

4. Some variables in some equations were not defined. It is not consistent.

5. Equation (15): (Yn)(Phi) = ins. Based on Equations (13) and (14), the sizes of Yn and ins are 3x3 and 3x1, respectively, meanwhile the size of Phi is 4x1. The multiplication in Equation (15) involves the matrix multiplication of Yn and Phi to yield the result ins. Could you please explain in more detail how it works?

6. You did mention that "the value of each resistor can be obtained by inverting the formula (16)" (refer to line 186). Please include the inverse equation as your Equation (17).

7. The way how you write nodes is not consistent. Please refer to lines 202 and 203 (without brackets) and lines 208 - 215 (with brackets). It is preferable to use the term "node" consistently before writing the number without brackets (e.g., "node 1"), instead of variations like "node (1)" or just "(1)" without the term "node".

8. Could you please provide definitions for Y(1:3,1:3), Y(2:4,2:4), Y(1 3 4, 1 3 4) and Y(1 2 4, 1 2 4) in lines 208 - 215? While it is understandable that you are referring to rows and columns. However, it is preferable to define these terms first or present them in a more detailed manner.

9. Please rewrite all the variables in Tables 1 - 10 in "equation" format.

10. Basically, this research paper is showing on how to derive a symmetric resistance network matrix by doing calibration with galvanic logging instruments. I think the title should include "calibration" and "galvanic logging instruments". Please consider this suggestion.

11. I believe that Equation (28) should not be included in the manuscript. The equation is related to aspects of the work that have not been explained in the manuscript.

12. Statements in lines 281 - 283 and 291 - 293 are repeated.

13. In the abstract, you mentioned that the symmetrical resistance network can provide rapid calculation and improve the accuracy. However, your research work lacks results or proof supporting these statements.

14. It is advisable to include an analysis or validation procedure in the manuscript. While the paper has demonstrated the calibration process leading to the development of a symmetric resistance network matrix, providing the significance of this matrix for galvanic logging applications in terms of accuracy or processing time would enhance the overall content in the manuscript.

15. All the best!!!

6. PLOS authors have the option to publish the peer review history of their article (what does this mean?). If published, this will include your full peer review and any attached files.

Reviewer #1: No

Reviewer #2: No

---

## [Author Response · Author response to Decision Letter 0]

14 Feb 2024

Dear Editor:

A rebuttal letter labeled 'Response to Reviewers' that responds to each point raised by the academic editor and reviewer(s) is uploaded as a separate file.

A marked-up copy of our manuscript that highlights changes made to the original version labeled 'Revised Manuscript with Track Changes' is uploaded as a separate file.

An unmarked version of our revised paper without tracked changes labeled 'Manuscript' is uploaded as a separate file.

To Reviewer #1: 

1. The sentences and grammar should be polished and modified by a native speaker. The current version is a little bit hard to understand;

Agreed. We have polished and modified the manuscript carefully.

2. It is better to give a workflow of the calibration method and describe the relationship between FEM and the resistance matrix method;

Agreed. A workflow is added and the relationship between FEM and the resistance matrix method is elaborated.

3. Please validate the accuracy of the new method in a layered medium.

Agreed. The accuracy of our method is validated through the comparison between different logging technologies in two real well.

To Reviewer #2:

 1. It was well-written. However, the abstract shall be improved. Please include the important result in the abstract.

Agreed. We modified the abstract and added the key result.

2. All the variables, e.g., RB0, RB1, k0, Rx0, is1, es, etc., must be rewritten. They should be written in "equation" format.

Agreed. All the variables have been written in "equation" format.

3. All the mathematical expressions also should be rewritten in "equation" format.

Agreed. They have been rewritten in "equation" format.

4. Some variables in some equations were not defined. It is not consistent.

Agreed. We examined all variables and defined those without definition.

5. Equation (15): (Yn)(Phi) = ins. Based on Equations (13) and (14), the sizes of Yn and ins are 3x3 and 3x1, respectively, meanwhile the size of Phi is 4x1. The multiplication in Equation (15) involves the matrix multiplication of Yn and Phi to yield the result ins. Could you please explain in more detail how it works?

Agreed. This is our carelessness. We corrected the mistake and explained these Equation in more detail.

6. You did mention that "the value of each resistor can be obtained by inverting the formula (16)" (refer to line 186). Please include the inverse equation as your Equation (17).

Agreed. We added the inverse equation in the manuscript.

7. The way how you write nodes is not consistent. Please refer to lines 202 and 203 (without brackets) and lines 208 - 215 (with brackets). It is preferable to use the term "node" consistently before writing the number without brackets (e.g., "node 1"), instead of variations like "node (1)" or just "(1)" without the term "node".

Agreed. We rewrote all the nodes.

8. Could you please provide definitions for Y(1:3,1:3), Y(2:4,2:4), Y(1 3 4, 1 3 4) and Y(1 2 4, 1 2 4) in lines 208 - 215? While it is understandable that you are referring to rows and columns. However, it is preferable to define these terms first or present them in a more detailed manner.

Agreed. We noted them so that they make sense.

9. Please rewrite all the variables in Tables 1 - 10 in "equation" format.

Agreed. All the variables in Tables were rewritten in "equation" format.

10. Basically, this research paper is showing on how to derive a symmetric resistance network matrix by doing calibration with galvanic logging instruments. I think the title should include "calibration" and "galvanic logging instruments". Please consider this suggestion.

Agreed. The title has been modified.

11. I believe that Equation (28) should not be included in the manuscript. The equation is related to aspects of the work that have not been explained in the manuscript.

Agreed. Equation (28) has been deleted from the manuscript.

12. Statements in lines 281 - 283 and 291 - 293 are repeated.

Agreed. We removed duplicate statements.

13. In the abstract, you mentioned that the symmetrical resistance network can provide rapid calculation and improve the accuracy. However, your research work lacks results or proof supporting these statements.

Agreed. The description is not accurate. We edited these statements. As the nodes become more and more, the calibration circuit design is also more difficult to be equal to the real formation. This method of the manuscript is proposed as a standard calibration method to avoid the complex calibration design difficulty.

14. It is advisable to include an analysis or validation procedure in the manuscript. While the paper has demonstrated the calibration process leading to the development of a symmetric resistance network matrix, providing the significance of this matrix for galvanic logging applications in terms of accuracy or processing time would enhance the overall content in the manuscript.

Agree. We added two validation test to prove that the method have advantages.

15. All the best!!!

I appreciated your review and valuable suggestions. Best wishes to you!

---

## [Decision Letter · Decision Letter 1]

5 Mar 2024

PONE-D-23-37666R1A standard calibration method based on symmetric resistance network matrix for galvanic logging instrumentsPLOS ONE

Dear Dr. ji,

Thank you for submitting your manuscript to PLOS ONE. After careful consideration, we feel that it has merit but does not fully meet PLOS ONE’s publication criteria as it currently stands. Therefore, we invite you to submit a revised version of the manuscript that addresses the points raised during the review process.

**Please have an expert to review the paper in terms of English language**

We look forward to receiving your revised manuscript.

Kind regards,

Khalil Abdelrazek Khalil, Ph.D.

Academic Editor

PLOS ONE

Journal Requirements:

Reviewers' comments:

Reviewer's Responses to Questions

**Comments to the Author**

1. If the authors have adequately addressed your comments raised in a previous round of review and you feel that this manuscript is now acceptable for publication, you may indicate that here to bypass the “Comments to the Author” section, enter your conflict of interest statement in the “Confidential to Editor” section, and submit your "Accept" recommendation.

Reviewer #1: All comments have been addressed

Reviewer #2: (No Response)

2. Is the manuscript technically sound, and do the data support the conclusions?

Reviewer #1: Yes

Reviewer #2: Yes

3. Has the statistical analysis been performed appropriately and rigorously? 

Reviewer #1: Yes

Reviewer #2: N/A

4. Have the authors made all data underlying the findings in their manuscript fully available?

Reviewer #1: Yes

Reviewer #2: Yes

5. Is the manuscript presented in an intelligible fashion and written in standard English?

Reviewer #1: Yes

Reviewer #2: Yes

6. Review Comments to the Author

Reviewer #1: The author has developed a standard calibration method to study the association between the DC logging instruments and real formations in this manuscript. Compared with last manuscripts, the author has addressed all those concerns. This is a much better manuscript which merit publication after a minor revision.

1. Please polish English writing. Some mistakes are existed. For example:

Table 2-9 ==> Tables 2-9.

2. All variables in the equation should be defined with their units added.

Reviewer #2: Dear Authors,

I have thoroughly reviewed the manuscript titled “A standard calibration method based on

symmetric resistance network matrix for galvanic logging instruments” and would like to

commend you on the comprehensive and meticulous manner in which you have addressed all

my queries and concerns. your careful attention to detail and the thoughtful revisions made have

significantly strengthened the overall quality of the paper. I would like to acknowledge that the

modifications you implemented have not only clarified several aspects of the research but have

also enhanced the overall coherence of the manuscript. Considering the thoroughness of your

revisions and the significant improvements made, I am pleased to recommend the acceptance

of your paper for publication. I believe that your research makes a valuable contribution to the

field, and I am confident that it will be well-received by the readership. Once again, thank you for

your diligence and commitment to advancing scientific knowledge. I look forward to seeing your

work published in PlosOne journal.

7. PLOS authors have the option to publish the peer review history of their article (what does this mean?). If published, this will include your full peer review and any attached files.

Reviewer #1: No

Reviewer #2: No

---

## [Author Response · Author response to Decision Letter 1]

21 Mar 2024

We have substantially revised our manuscript based on the recommendations of editors and reviewers. All of the figures are removed from the manuscript. In particular, our paper has been edited and polished by a native English speaker to improve grammar, flow, succintness, and readability. All the issues and concerns that were highlighted in the previous review have been addressed, and we wish to resubmit our paper for reconsideration.

To Reviewer #1: 

1. Please polish English writing. Some mistakes are existed. For example:

Table 2-9 ==> Tables 2-9.

Agreed. The manuscript has been polished and edited by a native English speaker.

2. All variables in the equation should be defined with their units added.

Agreed. We have added the units to all variables.

To Reviewer #2:

1.I have thoroughly reviewed the manuscript titled “A standard calibration method based on

symmetric resistance network matrix for galvanic logging instruments” and would like to

commend you on the comprehensive and meticulous manner in which you have addressed all

my queries and concerns. your careful attention to detail and the thoughtful revisions made have

significantly strengthened the overall quality of the paper. I would like to acknowledge that the

modifications you implemented have not only clarified several aspects of the research but have

also enhanced the overall coherence of the manuscript. Considering the thoroughness of your

revisions and the significant improvements made, I am pleased to recommend the acceptance

of your paper for publication. I believe that your research makes a valuable contribution to the

field, and I am confident that it will be well-received by the readership. Once again, thank you for

your diligence and commitment to advancing scientific knowledge. I look forward to seeing your

work published in PlosOne journal.

Thank you very much for your affirmation of the article. Wish you all the best!

---

## [Decision Letter · Decision Letter 2]

27 Mar 2024

A standard calibration method based on a symmetric resistance network matrix for galvanic logging instruments.

PONE-D-23-37666R2

Dear Dr. ji,

We’re pleased to inform you that your manuscript has been judged scientifically suitable for publication and will be formally accepted for publication once it meets all outstanding technical requirements.

Kind regards,

Khalil Abdelrazek Khalil, Ph.D.

Academic Editor

PLOS ONE

Additional Editor Comments (optional):

Reviewers' comments:

Reviewer's Responses to Questions

**Comments to the Author**

1. If the authors have adequately addressed your comments raised in a previous round of review and you feel that this manuscript is now acceptable for publication, you may indicate that here to bypass the “Comments to the Author” section, enter your conflict of interest statement in the “Confidential to Editor” section, and submit your "Accept" recommendation.

Reviewer #1: All comments have been addressed

2. Is the manuscript technically sound, and do the data support the conclusions?

Reviewer #1: Yes

3. Has the statistical analysis been performed appropriately and rigorously? 

Reviewer #1: Yes

4. Have the authors made all data underlying the findings in their manuscript fully available?

Reviewer #1: Yes

5. Is the manuscript presented in an intelligible fashion and written in standard English?

Reviewer #1: Yes

6. Review Comments to the Author

Reviewer #1: Good job. The authors have addressed all my concerns. I have no more comments. I would like to give a publish recommendation as is.

7. PLOS authors have the option to publish the peer review history of their article (what does this mean?). If published, this will include your full peer review and any attached files.

Reviewer #1: No

---

## [Editor Report · Acceptance letter]

1 Apr 2024

PONE-D-23-37666R2 

PLOS ONE

Dear Dr. ji, 

I'm pleased to inform you that your manuscript has been deemed suitable for publication in PLOS ONE. Congratulations! Your manuscript is now being handed over to our production team.

Kind regards, 

on behalf of

Dr. Khalil Abdelrazek Khalil 

Academic Editor

PLOS ONE